# Subsets of Eosinophils in Asthma, a Challenge for Precise Treatment

**DOI:** 10.3390/ijms24065716

**Published:** 2023-03-16

**Authors:** Jakub Novosad, Irena Krčmová, Ondřej Souček, Marcela Drahošová, Vratislav Sedlák, Martina Kulířová, Pavlína Králíčková

**Affiliations:** 1Institute of Clinical Immunology and Allergy, University Hospital, Skokolska 581, 500 05 Hradec Kralove, Czech Republic; 2Faculty of Medicine in Hradec Králové, Charles University in Prague, Simkova 807, 500 03 Hradec Kralove, Czech Republic; 3Department of Pneumology and Phthisiology, University Hospital, Skokolska 581, 500 05 Hradec Kralove, Czech Republic

**Keywords:** eosinophils, immunophenotype, asthma, biological therapy, biomarkers

## Abstract

The existence of eosinophils was documented histopathologically in the first half of the 19th century. However, the term “eosinophils” was first used by Paul Ehrlich in 1878. Since their discovery and description, their existence has been associated with asthma, allergies, and antihelminthic immunity. Eosinophils may also be responsible for various possible tissue pathologies in many eosinophil-associated diseases. Since the beginning of the 21st century, the understanding of the nature of this cell population has undergone a fundamental reassessment, and in 2010, J. J. Lee proposed the concept of “LIAR” (Local Immunity And/or Remodeling/Repair), underlining the extensive immunoregulatory functions of eosinophils in the context of health and disease. It soon became apparent that mature eosinophils (in line with previous morphological studies) are not structurally, functionally, or immunologically homogeneous cell populations. On the contrary, these cells form subtypes characterized by their further development, immunophenotype, sensitivity to growth factors, localization, role and fate in tissues, and contribution to the pathogenesis of various diseases, including asthma. The eosinophil subsets were recently characterized as resident (rEos) and inflammatory (iEos) eosinophils. During the last 20 years, the biological therapy of eosinophil diseases, including asthma, has been significantly revolutionized. Treatment management has been improved through the enhancement of treatment effectiveness and a decrease in the adverse events associated with the formerly ultimately used systemic corticosteroids. However, as we observed from real-life data, the global treatment efficacy is still far from optimal. A fundamental condition, “sine qua non”, for correct treatment management is a thorough evaluation of the inflammatory phenotype of the disease. We believe that a better understanding of eosinophils would lead to more precise diagnostics and classification of asthma subtypes, which could further improve treatment outcomes. The currently validated asthma biomarkers (eosinophil count, production of NO in exhaled breath, and IgE synthesis) are insufficient to unveil super-responders among all severe asthma patients and thus give only a blurred picture of the adepts for treatment. We propose an emerging approach consisting of a more precise characterization of pathogenic eosinophils in terms of the definition of their functional status or subset affiliation by flow cytometry. We believe that the effort to find new eosinophil-associated biomarkers and their rational use in treatment algorithms may ameliorate the response rate to biological therapy in patients with severe asthma.

## 1. Introduction

The existence of eosinophils was documented histopathologically in the first half of the 19th century in Gottlieb Gluge’s textbook of pathology from 1843. However, the term “eosinophils” was first used by Paul Ehrlich at the end of the 19th century (1878). At that time, their existence was associated with the pathogenesis of asthma. During the 20th century, the notion of the role of eosinophils in the human body was subject to considerable turbulence. While eosinophils were perceived as regulatory elements that dampen the proinflammatory activity of mediators released by mast cells during the 1970s and 1980s, in the later 1980s and 1990s, eosinophils began to be considered as terminal effector cells of immunopathological inflammation, causing tissue damage. The presence of eosinophils has traditionally been associated with defence against helminthic infections and with the pathogenesis of allergies. Since the beginning of the 21st century, theories about their physiological roles in the human body have been further revised. Furthermore, in 2010, J. J. Lee, the former president of the International Society for Eosinophils, proposed the concept of “LIAR” (Local Immunity And/or Remodelling/Repair) [1] in both health and disease. This impressive work has given rise to some controversies regarding the fundamental role of eosinophils in immune functions during the phylogeny, as follows: (1) Why have the host organisms evolved with a unique hematopoietic lineage as a defence mechanism against selected pathogens that are generally not life-threatening? (2) Why are the eosinophilic leukocytes absent from nearly all metazoans and present only among the five classes of vertebrates in the phylum Chordata (600 million years)? Furthermore, (3) if eosinophils were a prominent innate host defence against helminths across mammal species, why have pathogen-driven selective pressures not led to alternative or overlapping poietic pathways that could promote the expansion of these cells that would be independent of the eosinophils’ principal growth factor—interleukin-5 (IL-5)? [1].

Aside from the presumed antimicrobial functions, it has been suggested that eosinophils play roles in tissue morphogenesis, mucosal homeostasis, and metabolism, although this evidence has mainly come from preclinical models [2]. According to this theory, the regulatory and immunomodulating potential of eosinophils has been confirmed in the gastrointestinal tract (additionally in the small intestine, which represents the physiological reservoir for eosinophils in the body) [3], in adipose tissue [4], and in the lungs [5]. A common characteristic of tissues containing a high count of eosinophils is a high cellular turnover [1]. It has become apparent that mature eosinophils are not an immunologically homogeneous cell population but form more or less structurally and functionally different subtypes, which have separate development, localization, role and fate characteristics in tissues, especially during states of inflammation activation [5,6,7]. Therefore, a new classification scheme for eosinophil subsets has been proposed, which consists of eosinophil progenitors, steady-state eosinophils, and regulatory/resident eosinophils (rEos) and pro-inflammatory eosinophils (iEos) [5,8].

It has become evident that understanding such developmental phenotypes is inevitable. It can teach us about the origins and activities of functionally distinct eosinophilic cells during inflammation and explain the reasons for the successes and failures of biologics targeting this cell line.

## 2. Development, Phenotypes, and Functional Characteristics of the Eosinophil Lineage: Generation of Eosinophil Endotypes

In the current simplified paradigm, hematopoietic stem cells (HSC) from the bone marrow directly give rise to eosinophil/mast cell progenitors (EoMCP), from which eosinophil progenitors (EoPs) develop and terminally differentiate into mature eosinophils. EoMCPs also differentiate into basophil progenitors (BaP) and mast cells. Aside from EoMCPs, megakaryocyte/erythroid progenitors (MEP), common lymphoid progenitors (CLP), and common myeloid progenitors (CMP) are generated from the HSCs. In the forthcoming steps, the CMP is a source of the subsequent progenitors, neutrophil and monocyte progenitors (NMPs) and megakaryocyte/erythrocyte progenitors (MEPs), which give rise to particular haematological elements [9]. These progenitors and mature cells reveal a distinct immunophenotype, which enables their detection using flow cytometry [10] (see Figure 1 and Table 1).

Three central cytokines influence the maturation of eosinophils in the bone marrow—colony-stimulating factor for granulocytes and macrophages (GM-CSF), interleukin 3 (IL-3), and especially, IL-5 [11]. IL-5 also initiates the release of mature cells from the bone marrow and prolongs their survival in peripheral tissues [12]. Despite IL-5 being a key cytokine that regulates the generation and maturation of eosinophils, it is not entirely indispensable for eosinophil development, as IL-5-deficient mice and human patients receiving anti-IL-5 therapy all retain residual eosinophils [13]. Moreover, there is some level of redundancy in the IL-5 activity, since it can be produced by numerous cells (incl. Th2 lymphocytes, innate lymphoid cells (ILC)-2, NKT cells, and to a lesser extent, mast cells, epithelial cells, Reed–Sternberg cells, and EBV-transformed cells of eosinophils). It can also interact nonexclusively with eosinophils but also with basophils and possibly others [14,15].

Significant phenotypic changes occur during maturation, mainly caused by the gradual attenuation of the activity of the transcription factor FOG-1 (Friend of GATA-1) in the EoMP stage in favor of the actions of other transcription factors, GATA-1 and GATA-2 (zinc finger DNA binding proteins, that bind to the consensus DNA sequence (T/A)GATA(A/G) [16]) in EoP. Further, there is a transient increase in the activity of the transcription factors C/EBPε (CCAAT/enhancer binding protein) and finally ID-2 (DNA-binding protein inhibitor) in the mature eosinophil stage [17]. This switching leads to a gradual decrease in the expression of the surface molecule CD34 (adhesion phosphoglycoprotein specific for hematogenous stem cells) and, conversely, increases in the receptors for growth factors regulating their production and maturation, mainly the receptors for IL-3 (IL-3R) and GM-CSF (GM-CSFR) as well as the receptor for eotaxin-1 (CCR3). However, as the very first and functionally superior receptor, IL-5R is expressed on the surface of maturing eosinophils (a receptor for IL-5 consisting of the alpha subunit (IL-5Rα) and a common β chain—also described for the receptors for IL-3 and GM-CSF), the production of which is probably (at least in the EoP stage) positively regulated [18]. It has also been shown that the number of progenitors in the bone marrow (expressing CD34, CCR3, and IL-5Rα) increases 24 h after the exposure to an allergen in atopic individuals [19].

During its maturation in the bone marrow, in addition to the development of the surface receptor structure, under the influence of the transcription factors mentioned earlier (especially C/EBPε), specific (formerly secondary or α-granules) and primary (formerly β-granules, which are considered to be the immature form of specific granules) granules are gradually formed. There is also the formation of lipid bodies and so-called “sombrero vesicles” (organelles named based on their ultramicroscopy morphology). Each of these structures contains different biologically active substances, and their intentional release is a crucial element of eosinophil effector functions, both activating and inhibitory [20].

The complete surface receptor structure on the cell surface of mature cells enables the interaction of eosinophils with signals from the external environment, both activating (IL-3, GM-CSF and IL-5—through receptors IL-3R, GM-CSFR, IL-5R) and inhibitory (TGFβ—transforming growth factor β, via the TGFβR receptor or the Sialyl–Lewis X molecule (CD15s) via the Siglec-8 receptor—receptor family—Sialic acid ImmunoGlobulin-like LECtins), as well as with chemokines that attract eosinophils to the site of inflammation (eotaxin-1, 2, and 3, RANTES—Regulated upon Activation, Normal T Cell Expressed, and Presumably Secreted or PgD_2_ (prostaglandin D2) via CCR3, CCR1, and DP2/CRTh2 receptors, respectively). Their mutual balance represents an essential tool for controlling the production and migration of this cell population [21]. The surface immunophenotype of mature eosinophils is shown graphically in Figure 2 and Table 2.

However, immature progenitors of eosinophils can also leave the bone marrow and infiltrate the distant tissues (especially during the inflammatory reaction), allowing “in-situ eosinophilopoiesis”, which was also described in the mucous membrane of the respiratory tract [22] and may be associated with intensive local immune reactivity [23]. Moreover, mature eosinophils are heterogeneous cells with potentially multiple subsets. Only one subset of eosinophils is commonly detected in the blood and lungs of mice at a steady state. However, following the development of, e.g., an airway allergy to house dust mite antigens, at least two subsets of eosinophils become detectable in these animals. Similar subsets have also been detected in murine models of eosinophilic esophagitis [5]. These observations raise a question about the mechanisms leading to the endotyping of inflammatory eosinophils during eosinophilopoiesis or, instead, illustrate a picture of plasticity in the inflamed tissue. Since there are probably differences between the eosinophilopoiesis in health and in a disease state, it is likely that inflammatory signals originating from the site of inflammation, including IL-5, may modulate eosinophil development to impart functions on (subsets of) eosinophils that differ from those of their steady-state counterparts [2].

With better marker coverage by multicolor flow cytometry and single-cell level sequencing of granulocyte populations, novel phenotypes of these cells began to emerge. It is worth noting that many of these newly described subsets blend distinctions between classical myeloid lineage phenotypes. Aside from terminally differentiated eosinophils and other granulocytes (such as neutrophils) defined by conventional flow cytometry and granular protein markers, intermediate phenotypes with mixed neutrophil–eosinophil characteristics are coming to light. Two different processes may be responsible for the plasticity of granulocyte lineages: hematopoietic flexibility of granulocyte precursors and the adaptation of different subsets to local tissue and cytokine microenvironments [24].

## 3. Trafficking of Eosinophils from the Bone Marrow to the Peripheral Tissues

After leaving the bone marrow, the stay of eosinophils in the peripheral blood is relatively short (according to various sources, approx. 8–24 h). Unlike neutrophils, the circulating blood is primarily a transport medium for these cells [11]. Within 24 h, eosinophils reach their maximum levels in peripheral tissues (e.g., in the lungs, after stimulation with an allergen), where they remain for about eight days [25]. In addition, relatively high intraindividual circadian variability has been demonstrated in eosinophils in the peripheral blood, which is most likely related to the serum cortisol concentration [26]. Thus, endogenous glucocorticoid production is one of the most critical physiological regulatory mechanisms affecting the formation, activity, migration, and survival of eosinophils. This effect is particularly evident in all conditions involving increased cortisolemia, e.g., acute phases of viral and bacterial infections.

An essential condition for the entry of eosinophils into peripheral tissues is their adhesion to endothelial cells, which depends on the expression of adhesion molecules on both sides of mutual contact. This role is mediated both by the glycoprotein PSGL-1 (P-Selectin Glycoprotein Ligand 1, CD162) [11] and also by an extensive family of integrins [27]. Several integrin heterodimers expressed by eosinophils, such as very late antigen 4—VLA-4 (CD49d/CD29), complement receptors 3 and 4—CR3 (CD11b/CD18) and CR4 (CD11c/CD18), and lymphocyte function-associated antigen 1—LFA1 (CD11a/CD18) mediate diverse functions, including eosinophil rolling, stable adhesion, and migration and additional functions, such as respiratory bursts, degranulation, and viability after interaction with ligands including adhesion molecules, laminin, fibrinogen/fibrin, vitronectin, and periostin on other cells or in the extracellular matrix (ECM) [28]. Moreover, some selectins are found on the eosinophil surface, including P-selectin (CD62P) and L-selectin (CD62L) [29]. While eosinophils are mainly primed for adhesion by IL-5, endothelial cells are mostly orchestrated by IL-4 and IL-13 [11].

Eosinophils spend the largest and most significant part of their life in peripheral tissues (where most eosinophils are found even under physiological circumstances). Eosinophils are attracted there by chemotactic factors such as eotaxins 1, 2, and 3 [30] or the chemokine RANTES (Regulated on Activation, Normal T cell Expressed and Secreted) [31], especially during inflammatory reactions of the organism. Their presence may therefore be associated with organ impairment in these cases. In many tissues and organs (e.g., thymus, spleen, lymph nodes, bone marrow, uterus, mammary glands, adipose tissue, or the gastrointestinal tract), eosinophils occur completely physiologically, and their presence is not associated with any defined pathology [32,33], despite the described degranulation [34]. The migration of eosinophils is shown schematically in Figure 3.

## 4. The Conventional Concept of Eosinophils’ Role in Tissues

Studies conducted after 2010 mainly on animal models (especially mice) have repeatedly shown that tissue eosinophils contribute to the regeneration of muscle tissue, the normal development of the mammary glands and genitals, and the production of IgA antibodies in the digestive tract, as well as influencing the sensitivity of adipose tissue to insulin [3,4,5,7]. In the bone marrow, eosinophils also support B-lymphocyte maturation by producing IL-4 and IL-6. They keep plasma cells alive for a long time. Eosinophils probably also have an immunomodulatory role in forming the repertoire of specific T-lymphocyte receptors in the thymus as part of the central induction of tolerance [35]. Th2 immune responses and homeostatic eosinophil activities may have evolved not to expel parasites but to limit inflammation, control tissue glucose uptake, and minimize potential damage to the host [36]. There is evidence that some parasitic infections can paradoxically “abuse” the regulatory functions of eosinophils for their profit. Eosinophils, on this occasion, prevent the creation of an otherwise highly toxic microenvironment under the influence of interferon-gamma (IFNγ) and NO [37]. Indeed, it has been shown in murine models that the survival rate of the helminth *Trichinella spiralis* in the host organism is lower if the host organism cannot produce sufficient eosinophils [38].

The essence of the regulatory functions of eosinophils is their ability to release highly active biological substances formed in cell organelles into their surroundings. Eosinophils have a greater oxidative burst capacity than neutrophils [23]. They can release cytotoxic proteins from their specific granules (cationic proteins, such as eosinophil cationic protein (ECP, RNase3)), major basic proteins (MBP 1 and 2—representing approx. 50% of all produced cationic proteins), eosinophil peroxidase (EPX), eosinophil-derived neurotoxin (EDN, DNase2), and lipid mediators from lipid bodies (leukotrienes, prostaglandins, platelet-activating factor (PAF)). All of these products directly or indirectly prevent the formation, propagation, and persistence of inflammation. Eosinophils also secrete Charcot–Leyden crystals (otherwise known as Galectin 10) from primary granules (immature forms of specific granules) [39,40,41]. In addition, specific granules are a very potent source of large numbers of cytokines (e.g., IL-2, IL-3, IL-4, IL-6, IL-13, GM-CSF), chemokines (e.g., CCL3, CCL5 (RANTES), eotaxin-1 (CCL11), macrophage inflammatory protein (MIP-1α), monocyte chemoattractant protein (MCP-1) or IL-8), and growth factors (e.g., neuronal growth factor (NGF), platelet-derived growth factor (PDGF), vascular and endothelial growth factor (VEGF), endothelial growth factor (EGF)) [39]. The activation process itself is then associated, among other factors, with the binding of surface receptors for IgG and IgA with constant fragments of the respective immunoglobulins, as well as with the actions of cytokines IL-3, GM-CSF, and IL-5, chemokines CCL11 and CCL5 (RANTES), and platelet-activating factor (PAF) [42]. A summary of the top products secreted by eosinophils is given in Figure 2.

The release of all synthesized substances can occur in several ways: (1) by classical exocytosis (especially in the environment of a bacterial infection), (2) through compound exocytosis (excretion of several different products at the same time through a newly formed “fusion” cell compartment, often present in helminth infections), and (3) the most frequently occurring piecemeal degranulation, which is mainly mediated by the so-called “sombrero vesicles”, which represent unique vesiculotubular structures that probably serve mainly to transport the contents of specific granules to the surface of the eosinophil—their number increases during eosinophil activation. Piecemeal degranulation is present in all eosinophils involved in inflammatory processes, e.g., in allergies, and this process is similar to the release of neurotransmitters from nerve fibers. The final method of release of active mediators to their surroundings occurs during (4) cytolysis after the necrotic death of an eosinophil [20,41] which, together with piecemeal (gradual) degranulation, is the most relevant role of eosinophils in tissues. After the cytolysis of eosinophils, isolated granules remain in the tissues and express receptors for cytokines, chemokines, and cysteinyl leukotrienes on their surfaces. The interaction with appropriate cytokines leads to the subsequent release of the contents of the granules [35]. Free eosinophil granules (FEGs) probably play a notable role in asthma pathogenesis [43]. They may also be associated with the generation of autoantibodies for eosinophil-associated proteins, such as EPX [44,45,46].

Thanks to their extracellular killing capabilities (which are implicitly related to degranulation and protein secretion), eosinophils have traditionally been associated with defence against helminthic diseases. Their overproduction has often been described in this situation (with a maximum of 11–30 weeks after infestation), although there is some controversy associated with animal models [47]. There are doubts about the evolutionary biology involved [1].

However, several proteins produced by eosinophils also have properties that allow them to defend against viral, bacterial, and fungal infections [32]. Eosinophils are capable of phagocytosis and intracellular killing (producing free oxygen and nitrogen radicals, expressing inducible NO synthetase (iNOS), and generating NO which is toxic to intracellular parasites, including viruses [48]) and antigen presentation to helper T-lymphocytes [49]. Some cationic proteins produced by eosinophils possess RNAase activity, and these cells can create so-called “extracellular DNA traps” (Extracellular Eosinophil Traps—(EET)). In this specific process, mitochondrial DNA is released and subsequently cross-linked with the contribution of free oxygen radicals, cationic proteins (MBP and ECP), and Charcot–Leyden crystals outside the cell. Concurrently with EETs, intact membrane-bound granules have also been observed to be released from cells. Since EET formation occurs in parallel with eosinophil lysis, the term EETosis was coined, and cells were presumed to undergo necrotic death [50]. These EETs can eliminate gram-negative bacteria, e.g., in the gastrointestinal tract mucosa [51,52], and some parasites and viruses. They can also play an essential role in the pathogenesis of diseases such as asthma [50]. The list of antimicrobial functions of eosinophils should include their ability to interact with T-helper cells as antigen-presenting cells and modify their cytokine production [53]. They may induce interferon production by epithelial cells as well [54]. Finally, the measurement of eosinophil activation products (especially ECP) in the peripheral blood has become routine for allergy diagnostics [55]. A summary of the mechanisms involved in the antimicrobial defence of eosinophils is presented in Figure 4 and Table 3.

Despite the controversy regarding the proper biological roles of eosinophils in phylogeny, more recently, especially in the context of the COVID-19 pandemic, the putative role of eosinophils in antiviral protection has gained some momentum [2,56]. In addition, eosinophil depletion with anti-IL-5Rα therapy seems to be associated with an increased risk of bacterial-infection-induced asthma exacerbation [57].

The tissues and immunopathological activity of eosinophils can also have several clinically risky manifestations or patterns: (1) the formation of thrombi (through the release of tissue factor, MBP, and EPX) [58], (2) neuronal damage (mainly through EDN, which is associated with the inhibition of muscarinic M2 receptors by the actions of MBP), and finally, (3) tissue fibrotization (stimulated by the actions of ECP, MBP, TGFβ, and IL-1) [49]. If the activities of eosinophils occur in the environment of the lung tissue and airways, it would not be surprising that eosinophils play a key role in remodelling processes [59], as shown in Figure 5.

## 5. Eosinophils as a Biomarker in Asthma

The general examination of eosinophils in patients with asthma has a unique position among other biomarker measurements, based mainly on the convincing evidence of their crucial role in the pathogenesis of the disease itself. Their presence in the sputum initiated the first works dealing with asthma’s variable susceptibility to corticosteroid therapy in the 1950s. Their findings in biopsy and cytological samples of the airways inspired the concept of eosinophilic and noneosinophilic asthma, eventually giving rise to the terms phenotype and endotype. As already mentioned above, currently, the inflammatory endotypes of asthma are referred to as “type 2-high” and “type 2-low”, where the primary discriminating factor is the presence of eosinophilic inflammation (more recently referred to as type 2 inflammation). If signs of type 2 inflammation are present, we must distinguish whether it occurs primarily based on specific allergic, atopic reactivity or whether the inflammatory process is controlled by nonspecific, nonallergic, nonatopic mechanisms through innate immunity.

The atopic reactivity is under the guidance of the helper T lymphocyte subset 2 (Th2, a subtype of asthma known as “Th2-high”) and is dependent on the interaction of specific T cell receptors (TcR) with an allergen fragment presented by a dendritic cell. In contrast, nonspecific immune reactions or innate immune reactions occur independently of specific antigen recognition. The innate immune response is guided by innate lymphoid cell type 2 (ILC-2); therefore, we refer to this inflammatory endotype as “ILC2-high”. The immunological ramifications of both respective activation models are very similar, and eosinophils as present in the core of the functional overlap [60,61]. The practical use of this immune–pathological concept of asthma inflammation led to the development an immune–phenotype-based approach for asthma diagnostics and classification [62]. A schematic representation of the mechanisms of type 2-high inflammatory activity is shown in Figure 6.

The identification of the inflammatory endotype of asthma is necessary not only for diagnosing the disease but also for estimating its clinical severity and burden [63], the risk of course deterioration, exacerbations, bronchial hyperreactivity, and airway remodelling [64,65,66,67]. It is also a significant predictor of the response to the conventional anti-inflammatory treatment with inhaled or systemic corticosteroids [68,69]. It can also be used to assess compliance and adherence to treatment [70].

Currently, we use a triplet of validated and fully standardised biomarkers to detect the Type 2-high inflammatory endotype in asthma patients—the absolute eosinophil count in the peripheral blood (AEC), a fraction of nitric oxide in the exhaled breath (FeNO), and the production of immunoglobulin E (IgE) [71,72]. Nonetheless, conventionally, a relative number of eosinophils in induced sputum exceeding 2 or 3% is the diagnostic gold standard of the eosinophilic inflammatory phenotype. This characteristic is not universal. We can also evaluate the number of eosinophils in broncho-alveolar lavage fluid (BALF) or directly through a biopsy of the airway mucosa for this purpose. However, the most important data are obtained by examining induced sputum. However, these measurements may be burdensome for the patient and are demanding in terms of obtaining the materials and carrying out their subsequent laboratory processing. The apparent consequence has been an effort to find an alternative way to estimate or even affordably replace this parameter. The first rational option is the examination of eosinophils in the peripheral blood [73]. A comparison of the cut-off values used in particular methods to detect pathological eosinophilic inflammation in asthma is summarized in Figure 7.

Eosinophils in induced sputum (or in a biopsy specimen or BALF) and peripheral blood show significant variability, both intra- and interindividually, resulting mainly from physiological changes in the cortisolemia [26] due to the activity of the disease or as an effect of the therapy (both systemic and local). Several studies have shown that the examination of peripheral blood eosinophils (after exceeding the critical value) can predict the induction of sputum eosinophilia with all of the diagnostic and therapeutic consequences, including the sensitivity of the disease to treatment with inhaled steroids. However, other papers did not confirm this assumption. Some researchers have stated that no conventionally used biomarker in peripheral blood (eosinophils or IgE) or exhaled air (FeNO) can be used to reliably estimate the presence of eosinophils in sputum [74].

Undoubtedly, an immense amount of information and data on the inter-relationships between blood and tissue eosinophils (or eosinophils in induced sputum) has been obtained thanks to biomarker and efficacy studies that have primarily focused on the use of biological therapies directed against IL-5. The first randomized controlled trial (phase IIa) assessed the effect of a single intravenous infusion of the first anti-IL-5 monoclonal antibody [SB-240563] on the airway status after a challenge with an inhaled allergen in unselected patients with mild asthma conducted in 2000. However, the result was a significant reduction of eosinophils in the peripheral blood for 16 weeks and the induction of sputum four weeks after infusion, with no effect on clinical outcomes—late allergic reaction or histamine-induced hyper-reactivity [75]. Disappointment with the failure of antieosinophilic therapy in asthma has even led to doubts about the causal link between eosinophils and asthma. However, the fundamental reason for this failure was the significant biological variability of the disease, which alleviated the likelihood of achieving statistical significance in clinical treatment outcomes. It was, therefore, necessary to better specify the target group of asthmatics who could benefit from the therapy. In 2008, Haldar conducted his first cluster analysis, which demonstrated, to a certain extent, the possible independence of clinical manifestations on the intensity of inflammation and thus the existence of discordant forms of the disease (symptoms without inflammation or, conversely, inflammation without symptoms) [76]. When study groups applied the treatment only to asthmatics with overwhelming eosinophilic inflammation, defined as sputum eosinophilia despite systemic corticosteroid therapy, statistically and clinically significant effects were achieved [13,77,78]. Initially, these patients were characterized by increased eosinophils in the induced sputum (≥2–3%). Later (to enable a routine evaluation), they were classified by an increase in eosinophils in the peripheral blood (≥150/μL at the time of screening or ≥300/μL at least once in the last year) [79]. The main contribution of these observations was the definitive proof of the effectiveness of antieosinophilic anti-IL-5 treatment in patients with asthma and the appreciation that identifying the inflammatory endo-phenotype of asthma (either by examination of the induced sputum or peripheral blood) is an indispensable requirement for treatment success.

In the subsequent phases of clinical biological testing directed against IL-5, based on examination of the absolute number of eosinophils in the peripheral blood, threshold values of the absolute count of peripheral eosinophils have most often been set in the range of 150–400/μL. However, it can generally be assumed that the higher the number of eosinophils in peripheral blood is, the greater the probability their presence in tissues, including induced sputum, is. At the same time, the effect of antieosinophil treatment is greater [80]. It is also worth noting that the relationship between eosinophils in peripheral blood and induced sputum also depends on the specific type of disease present, as shown by a regression analysis of the relationship between both variables in patients with asthma and chronic obstructive pulmonary disease (COPD) [81].

Regarding the discussion of the usefulness of examining peripheral eosinophils to estimate airway eosinophilia and thereby identify the inflammatory endotype, the risk of worsening or exacerbating asthma and the evaluation of its responsiveness to therapy (biological or conventional anti-inflammatory) continues to persist. Manali Mukherjee and Parameswaran Nair highlighted many issues: (1) the concentration of eosinophils in the induced sputum can predict the loss of asthma control with higher sensitivity compared to their concentration in the peripheral blood [82], (2) the correlation between eosinophils in the peripheral blood and the induced sputum decreases as the severity of asthma increases, (3) eosinophils in the peripheral blood do not necessarily reflect the phenomenon of peripheral tissue eosinophilopoiesis, (4) an increase in the number of eosinophils in the peripheral blood has been demonstrated, for example, in connection with therapy with lebrikizumab (anti-IL-13) or dupilumab (anti-IL-4Rα) without causing the loss of asthma control [83,84]), and (5) rather than using their numbers alone, it is necessary to take into account the functional state, activation, and degranulation of these cells [85].

There are, however, even more unsolved questions and associated doubts regarding the reliable implementation of peripheral blood eosinophilia examination into routine clinical practice. These include (6) uncertainty regarding the association of clinical outcomes with the values of peripheral eosinophilia, (7) determination of the genuinely reliable threshold value (if one exists) of blood eosinophilia for predicting the clinical course of the disease and its response to targeted therapies directed against IL-5(R), (8) determination of the number of times it be measured (due to its variability), and finally, (9) whether monitoring of the number of eosinophils in the peripheral blood can be used to monitor the treatment effectiveness, especially in patients with a severe form of the disease, in whom a low degree of correlation with eosinophils in the induced sputum has been proven [80].

In summary, the examination of eosinophils in the peripheral blood is necessary, although still debatable and problematic, when reviewing a patient with asthma. Undoubtedly, it provides information regarding the inflammatory endotype, the estimation of the prognosis, and the probability of the effect of anti-inflammatory treatment (steroids or biological therapy, mainly directed against eosinophils). The recommended cut-off values for the absolute number of eosinophils in patients with asthma, their frequency, and interpretation are summarized in Table 4 [86,87,88].

## 6. Anti-Eosinophil Targeted Therapy in Asthma

Asthma treatment is principally directed against two key pathogenetic pillars—bronchial obstruction and airway inflammation. The treatment of inflammation with inhaled corticosteroids has proven to be crucial and has been the dominant recommendation for all disease severities since the 1970s. Unfortunately, more than 2–3% of asthmatics suffer from disease that is refractory to treatment with a tendency to have severe exacerbations and a necessity for repeated or continuous treatment with systemic corticosteroids, which come with complications and consequences [72,89,90]. Limitation of the side effects associated with systemic corticosteroid therapy has led to the development of selective drugs based on monoclonal antibodies directed against key immunopathogenic processes. These drugs show greater specificity and safety and have brought about a revolution in treatment. Since 2003 (when the first drug was registered by the American Food and Drugs Administration—FDA), a therapy enabling the inactivation of the IgE molecule (with the monoclonal antibody omalizumab) has been in clinical use. After more than ten years, therapeutic monoclonal antibodies directed against IL-5 mepolizumab (anti-IL-5) (2015), reslizumab (anti-IL-5) (2016), and benralizumab (anti-IL-5Rα) (2017) were approved. Two years later, the monoclonal antibody dupilumab (anti-IL-4Rα) (2019) and, finally, tezepelumab (anti-TSLP) (2022) were registered as well. Since 2015, in the therapeutic algorithm of the Global Initiative for Asthma (GINA), biological treatment has been indicated even before the start of systemic corticosteroid therapy. This critical guideline renewal significantly improved the safety of treating patients suffering from a severe form of asthma who would be otherwise seriously harmed by systemic steroids [91]. The current recommendation for appropriate biological treatment selection depends on the natural characteristics defined by the evaluation of biomarkers (AEC, atopic reactivity, IgE production, and measurement of FeNO [71,72]. However, substantial overlaps exist. It is, therefore, difficult to recommend only one biological therapy for a particular patient with defined biomarker characteristics. For example, in one exploratory analysis, 24% of 137 analyzed severe asthma patients concurrently fulfilled the eligibility criteria for all (that time) available biologics (mepolizumab, benralizumab, and omalizumab) [92].

The following text focuses on registered drugs that are primarily directed against eosinophils (mepolizumab and reslizumab–anti-IL-5 and benralizumab–anti-IL-5Rα) and drugs that are currently being researched (depemokimab–anti-IL-5 and lirentelimab–anti-Siglec-8).

The monoclonal antibody mepolizumab was used for the first time in an animal model in 1992 [93]. Less than eight years later, the first human trials were launched with the first available preparation (SB-240563, later mepolizumab, a humanized monoclonal antibody against IL-5, IgG_1_, κ). The registration studies DREAM (iv mepolizumab in patients with severe refractory eosinophilic asthma defined by an increased number of eosinophils in the sputum, ≥3%) [94] and MENSA (iv mepolizumab vs. sc mepolizumab in patients with severe refractory eosinophilic asthma defined by an increased number of eosinophils in the peripheral blood, ≥150/μL or at least once in the previous year ≥300/μL) [79] initiated the approval of mepolizumab for the treatment of asthma. These studies were followed by another, which investigated the sparing effect of mepolizumab on steroid treatment—SIRIUS (sc mepolizumab in patients with severe refractory eosinophilic asthma defined by an increased number of eosinophils in the peripheral blood) [95]. Finally, the MUSCA study [96] focused on patients’ quality of life in terms of their health status as well on patients with severe refractory eosinophilic asthma defined as an increased number of eosinophils in the peripheral blood). The DREAM study was subsequently expanded in the open-label follow-up phase IIIb (COLUMBA) [97], and the MENSA and SIRIUS studies were continued by the open-label observational phase IIIb COSMOS study [98], followed by the COSMEX (NCT02135692) and OSMO (NCT02654145) studies [99]. All mentioned extensions were intended to monitor the safety profile and efficacy of mepolizumab over a more extended period of time. In contrast to other antieosinophilic biologics, mepolizumab is approved for use in other conditions, namely chronic rhinosinusitis with nasal polyps (CRSwNP) [100], eosinophilic granulomatosis with polyangiitis (EGPA) [101,102], and FIP1L1-PDGFRA-negative hypereosinophilic syndrome (HES) [103].

Reslizumab (SCH55700) is a humanized monoclonal antibody (IgG_4_, κ) that, similarly to mepolizumab, is directed against human interleukin-5 (IL-5). Its binding prevents its interaction with the relevant receptors on the cell surface. The development of reslizumab followed the academic and clinical mepolizumab programs. Therefore, when reslizumab entered the market, it was already clear that specific biological therapies can only achieve a clinically relevant effect in selected groups of patients.

Since 2003, the following studies have been published: a pilot study, a phase II study, and 4 phase III studies. The clinical benefits of reslizumab were demonstrated as a reduction in the frequency of severe exacerbations and an improved lung function. This result was visible in patients with eosinophilic asthma, defined as an absolute number of eosinophils in the peripheral blood of ≥400/μL [104,105] or by an increase in the representation of eosinophils in the induced sputum of ≥3% [106] and in patients with concomitant chronic rhinosinusitis with nasal polyps [107]. All included subjects had experienced at least one severe exacerbation in the previous year. The studies showed that, in patients who responded well to treatment, there was an increase in the quality of life and a reduction in the consumption of systemically used corticosteroids and the side effects associated with the therapy. The treatment was also shown to have a very high safety level [108,109]. The disease-modifying effect (or effect on possible airway remodelling) of reslizumab is unclear [110].

Benralizumab (MEDI-563) is a humanized, afucosylated, monoclonal antibody (IgG_1_, κ) that binds to the alpha subunit of the IL-5 receptor (IL-5Rα) on the surface of eosinophils. The emergence and development of benralizumab were intrinsically linked to the description of the structure and function of IL-5R, which is expressed explicitly on the surfaces of eosinophil precursors and basophils. In addition, the absence of fucose on the Fc fragment of benralizumab enables high-affinity binding to FcγRIII receptors on immune effector cells, such as natural killer (NK) cells, thus facilitating the induction of the apoptotic death of eosinophils and basophils by antibody–dependent cell-mediated cytotoxicity (ADCC) [111]. Combining both effects causes the rapid depletion of eosinophils and eosinophil precursors (unlike the earlier biologics that were directed only against IL-5) within 24 h after the first drug administration. Elimination (of mature eosinophils and eosinophil precursors) has been achieved in the respiratory tract, peripheral blood, and bone marrow [112].

Moreover, thanks to its regulated process, the apoptotic death of eosinophils prevents the uncontrolled release of the content of cytotoxic granules. Therefore there is no paradoxical rise in the concentration of IL-5 [113,114]. During the first phases of clinical trials, benralizumab, like other biologics, was shown to have a favorable safety profile [115].

The efficacy of benralizumab was assessed in larger cohorts in three registrational, randomized, double-blind, placebo-controlled phase III clinical trials. The SIROCCO [116] and CALIMA [117] trials included patients with severe, uncontrolled asthma and a history of 2 or more severe asthma exacerbations requiring oral or systemic corticosteroid therapy (mean 3) in the past 12 months. They also reported a reduced baseline lung function before bronchodilation, despite regular treatment with high-dose inhaled corticosteroids (ICS) (SIROCCO) or medium or high-dose ICS and long-acting β agonists (LABA) (CALIMA). There was no threshold value for the number of eosinophils in the peripheral blood as a condition for inclusion in the study (unlike the previous preparations). However, in both studies, there was a significant reduction in the frequency of exacerbations, regardless of their value. However, a higher baseline value (≥300 cells/μL) was identified as a potential predictor of a better response to treatment in a posthoc analysis [118]. In addition, in patients with eosinophil counts ≥ 300 cells/μL, a significant improvement in FEV_1_ compared to the baseline level was recorded as early as week 4, which persisted until the end of the treatment, an effect that was generally positively correlated with the baseline eosinophil level [119].

Another posthoc analysis demonstrated the more pronounced effect of benralizumab on the reduction of exacerbations in the population of asthmatics whose initial eosinophil concentration did not reach 300/μL in the peripheral blood if they (1) simultaneously used oral corticosteroids, (2) suffered from nasal polyps, or (3) had a reduced forced vital capacity (FVC) <65% [120]. Finally, a more significant effect of benralizumab was also demonstrated if the value of eosinophils exceeded 150 cells/μL [118]. Similar to the case of mepolizumab, the impact of benralizumab was not dependent on the baseline level of IgE or the presence of atopic reactivity [121].

In the following study, ZONDA [122] asthma patients treated with systemic corticosteroids (median daily dose: 10 mg of prednisone equivalent) together with high doses of ICS and LABA who had ≥150 eosinophils/μL and with a history of at least one exacerbation were included. The study showed that subjects treated with benralizumab could reduce the dose of oral corticosteroids by a median level of 75%. In all of the studies mentioned above, adverse effects associated with benralizumab therapy were also carefully monitored and were comparable to those of a placebo. The side effects were generally not severe and were dominated by nasopharyngitis, the worsening of asthma, and bronchitis [123]. Patients with mild to moderate asthma were randomized in the BISE study (*n* = 211). There was an improvement in the prebronchodilation value of FEV_1_ after 12 weeks of treatment with benralizumab at a dose of 30 mg sc every four weeks [124]. In another “open-label” GREGALE trial, 116 patients with severe asthma were successfully treated at home with prefilled syringes [125]. The registration studies were followed by several other post-marketing follow-up studies that focused on both the effectiveness of the treatment and its safety. The SIROCCO, CALIMA, and ZONDA studies were extended by 56–108 weeks (total treatment time max. 164 weeks) in the BORA study [126]. This extension confirmed the treatment’s high levels of safety and tolerability. It was followed by a two-year extension study of patients randomized as part of the ZONDA study and the MELTEMI study (NCT02808819), the results of which are not yet available. A summary of the indication criteria concerning the number of eosinophils and other characteristics of included patients for all approved biologics (mepolizumab, reslizumab, and benralizumab) is given in Table 5. The effects of individually registered preparations (effects on eosinophils in the bone marrow, peripheral blood, and in the induced sputum) and the effects on surface receptors of eosinophils and other cells and cytokines are summarized in Table 6 [14,15,127].

In addition to the registered preparations that target eosinophils, we address two other drugs. The first is depemokimab (GSK3511294, a humanized monoclonal antibody (IgG_1_, κ) directed against IL-5 that interacts with the same epitope of IL-5 as mepolizumab but with a higher affinity and longer biological half-life [128]. This molecule is currently being tested in phase III clinical studies, SWIFT1 and SWIFT2 (NCT04719832, NCT04718103), as well as the NIMBLE study (NCT04718389), which is focused on the effectiveness of therapy switching from mepolizumab/benralizumab to depemokimab. Unlike the registered molecules, this antibody has a prolonged biological half-life, which makes it possible to apply it in a dose of 100 mg sc for 26 weeks.

The last selective antieosinophilic biologic drug is lirentelimab (AK002), a humanized, non-fucosylated IgG_1_ monoclonal antibody directed against the surface molecule of eosinophils, sialic acid–binding immunoglobulin-like lectin 8 (Siglec-8), an inhibitory receptor that is selectively expressed on mature eosinophils and mast cells and has a low level of expression on basophils [129,130]. Lirentelimab causes a depletion of eosinophils through natural-killer-cell-mediated antibody-dependent cellular cytotoxicity (ADCC in the blood) and apoptosis (in tissues). Moreover, lirentelimab inhibits mast cell activation, thereby reducing degranulation, the secretion of inflammatory mediators, and the recruitment of additional mast cells, eosinophils, and other immune cells to the tissues. Open-label clinical studies of lirentelimab have proved its activity in various allergic diseases, such as chronic urticaria (Clinical-Trials.gov number, NCT03436797) and severe allergic conjunctivitis (NCT03379311) [131]. Recently, in patients with eosinophilic gastritis or duodenitis, lirentelimab was shown to reduce gastrointestinal eosinophils and their symptoms [132]. It can be assumed that clinical studies on the effect of this drug on asthma will follow. The differences in the mechanisms of action between individual biologics directed against eosinophils are summarized in Figure 8 [127,133,134].

## 7. Intratissue Kinetics and Functional Development of Eosinophils, the Novel Concept of Eosinophil Functions and Plasticity

Eosinophils are an extraordinarily dynamic and inhomogeneous cell population at both the level of progenitors (mainly in the bone marrow, but per in-situ eosinophilopoiesis also in peripheral tissues) and at the stage of fully matured and immunocompetent cells and under both physiological and pathological circumstances, including in asthma patients. Indeed, it has historically been known that there are two different specific gravity cell types of eosinophils: those with a normal density (specific gravity > 1.085 g/L) and those with a lower density (specific gravity < 1.085 g/L), the latter of which represents the activated phenotype [135,136]. Prin et al. were the first to notice greater cytotoxicity in hypodense eosinophils, and these were also associated with a higher level of vacuolization [135]. Further research revealed that a higher count of hypodense eosinophils in peripheral blood is associated with the clinical severity of asthma and airway hyperreactivity [137]. These cells were also shown to be elevated in broncho-alveolar lavage fluid (BALF) following an antigen challenge [136].

Later, Abdala Valencia et al. analyzed the phenotype plasticity of murine eosinophils in digested lung tissue and BALF using flow cytometry 6 h after the ovalbumin challenge. The gradual transition from a Siglec-F^med^CD11c^−^ to a Siglec-F^high^CD11c^low^ phenotype in lung tissue was associated with eosinophil recruitment to the airways, as all BALF eosinophils were of the latter phenotype. Moreover, specific morphological differences were related to particular immunophenotypes (Siglec-F^med^CD11c^−^ eosinophils were typically ring-shaped, possible analoguous to normodense eosinophils, in contrast to Siglec-F^high^CD11c^low^ cells, which contained segmented nuclei, had higher vacuolization, and were less dense). The authors proposed that the Siglec-F^high^CD11c^low^ phenotype switch is associated with the migration of eosinophils into the mucosal compartment [138]. Similar eosinophil subpopulations have been described by others in allergen-challenged mouse lungs [139]. This concept has been further extended in the tissue-based classification of murine eosinophils differentiating between (1) eosinophil progenitors (immature eosinophils recruited as precursors or undergoing in situ hematopoiesis), (2) steady-state, and (3) type-1 eosinophils (corresponding to ring-shaped nuclei containing normodense Siglec-F^med^CD11c^−^), and (4) type-2 eosinophils (bearing the F^high^CD11c^low^ immunophenotype with segmented nuclei, vacuolization, and a lower density). Type-1 eosinophils are interstitial/stromal eosinophils that are found in transient morphogenetic contexts and during Type-1 (Type 2-low) immune activation. This is in contrast to Type-2 eosinophils, which are associated with the epithelium and Type-2 immune environment [8].

In accordance with this model of eosinophil functional divergence, in 2016, Mesnil et al. discovered phenotypically distinct subtypes of eosinophils in the murine lungs under steady-state conditions and after house-dust-mite (HDM)-induced allergic inflammation. These cells were classified as resident eosinophils, rEos (Siglec-F^med^CD62L^+^CD101^low^, IL-5 independent parenchymal cells with a ring-shaped nucleus), and inflammatory eosinophils, iEos (Siglec-F^high^ CD62L^−^ CD101^high^, which are IL-5 dependent peribronchial cells with a segmented nucleus). The study also demonstrated that rEos express genes that have been implicated in the negative regulation of the immune response, surprisingly inhibiting Type 2 inflammation. Additionally, the authors described two phenotypically distinct eosinophil subsets in the human lungs of healthy donors, CD62L^+^IL-3R^low^, and in the sputum of patients with asthma, CD62L^low^IL-3R^high^ [5]. Replication of the protocol unravelled the presence of these cells, even in the nasal polyps of patients suffering from CRSwNP [140]. Moreover, the classification based on morphological changes, density, or immunophenotype is almost interchangeable [27,141,142]. To summarize the previous statements, we have sound evidence that after the escape of eosinophils from the bloodstream, the tissue entrance is associated with a substantial morphological and functional transformation of mature cells and may be an example of intratissue trafficking. These immunophenotypic changes correlate with the disease severity [27] and reflect the activation of eosinophils in the tissues. For this purpose, we can follow many surface markers with positive (e.g., CD11b, CD11c, CD13, CD18, CD25, CD69, CD123 (IL-3Ra), HLA-DR or TSLPR) or negative (e.g., CR3, IL-5Rα and CD62L (L-selectin)) correlations with the eosinophil activation status [27,140,141,143] to differentiate between naïve and activated cells.

A considerable effort has been dedicated to unravelling the roles of distinct eosinophil subpopulations (especially rEos and iEos) [35]. In physiological situations, the rEos probably have a significant contribution to tissue regeneration [144,145] and possess a surprising potential to inhibit even Type 2 inflammation in the lungs [5]. On the contrary, the iEos cells have a potentially significant destructive force, leading to a possible tissue pathology [1,7,35]. Moreover, in line with the tissue kinetics of naïve and activated eosinophils, the rEos are located mainly in the lung parenchyma. This is in contrast to the iEos, which reside primarily in the airway mucosa and evade the sputum [2,5]. An extensive analysis of airway eosinophils supported this point of view while showing that intraepithelial eosinophils are associated with endogenous airway hyperreactivity (AHR) and Type 2 inflammation and may interact with intraepithelial mast cells via cysteinyl leukotrienes (CysLTs) to regulate airway inflammation [146].

Concurrently, whether this differentiation is initiated after entering the tissue microenvironment or may be detected even before in circulation has been debated. Alternatively, it is unclear whether these differences could arise through eosinophil “endotyping” as a pattern of adaptations imprinted during their development in the bone marrow in the peripheral circulation or through tissue-induced plasticity and local transformation of eosinophil function through interactions with their lung tissue niches [2].

Early studies in the 1980s reported the presence of hypodense eosinophils in the circulation of asthmatic patients [147], and these cells were initially interpreted as activated eosinophils. These cells display increased survival, adhesion, oxygen metabolism, superoxide production, and antibody-dependent cytotoxicity compared with normodense eosinophils [148]. Later immunophenotyping studies confirmed the presence of different subsets of eosinophils (rEos and iEos) in the peripheral blood and tissues [5,149]. However, it is still unclear whether the source of signals influencing eosinopoiesis originates from the bone marrow or the peripheral tissues [2]. Van Hulst et al. compared the gene expression profiles of circulating blood eosinophils in healthy controls and allergic or severe eosinophilic asthma patients (Type 2-high inflammatory endotype) treated either by mepolizumab (anti-IL-5) or omalizumab (anti-IgE). They observed that there was virtually no difference in the gene expression program of residual eosinophils from mepolizumab-treated patients compared to the eosinophils of the healthy controls and omalizumab-treated patients at steady state. These results, therefore, support the idea that eosinophil endotyping does not occur in asthma and that anti-IL-5 biologicals do not act by modulating this process [150]. Forthcoming studies should, therefore, assess and compare the heterogeneity of blood (and possibly bone marrow) eosinophils in asthma (or other eosinophilic disorders) patients with different phenotypes [2]. Finally, the mutual relationship between maturing tissue eosinophils and specific tissue microenvironments may be bilateral. Not only eosinophils but also the fully differentiated epithelial cells may undergo some reversible phenotypic transformation orchestrated by cytokines and induced by eosinophils (IL-4 and IL-13 or growth factors, e.g., TGFβ or fibroblast growth factor/FGF/)—the epithelial–mesenchymal transition (EMT) [151]. The current view of eosinophil subsets in tissue pathology is summarized in Figure 9.

## 8. What Questions Regarding Eosinophils in Asthma Remain to Be Solved

For over one hundred years, we have recognized that there is a clinically relevant connection between these cells and eosinophilic (Type 2-high) asthma, leading to advanced diagnostic and therapeutic opportunities. However, despite the enormous progress in this field, we are still struggling with central questions regarding the disease phenotyping, selection, timing, monitoring, switching, or quitting of specific antieosinophil biological treatments. Future questions to be answered include (1) how can we improve asthma diagnostics, especially regarding inflammatory endotype establishment (the precise role of biomarkers in immune-phenotype-endotype evaluation is still under debate)? (2) Which biological treatment will be the best for a particular patient (the immunological characteristics of potential responders for respective treatment choices overlap significantly)? (3) How can we monitor the treatment effect (is it more appropriate to follow clinical or laboratory measurements and what is the hierarchy of particular examinations and reliable interpretations)? (4) If we register the clinical effect of the treatment, how can we be sure that the efficacy is the best achievable? How fast and stable can the clinical response be? (5) Is the clinical response sufficient to continue the treatment, or should we consider switching or quitting (or maybe doubling) the treatment due to a lack of efficacy?

In everyday clinical practice, we must follow only validated and standardized biomarkers with well-defined cut-off levels for use in the diagnosis, treatment strategy, and monitoring of asthma patients (AEC or eosinophil count in the sputum, FeNO and allergological examination, including skin prick testing and assessment of total and specific IgE). All of these measurements aim to detect the Type 2-high inflammatory endotype and are not perfectly correlated due to differences in background cytokine regulation. Therefore, respective or mixed positivity indicates another signal in the disease pathology or a slightly different inflammation phenotype, thus supporting the use of a different treatment strategy [152]. It is, therefore, a frequent issue that one patient may be eligible for more therapeutic approaches. For example, 88% of all severe asthma patients included in the International Severe Asthma Registry (ISAR) had at least one positive biomarker, 59% had double positivity, and 27% were triple positive (cut-offs were AEC ≥ 300/µL, FeNO ≥ 25 ppb and IgE ≥ 75 kU/l) [153]. In another cross-sectional baseline data analysis of the general asthma population, 78% of all patients had the characteristics of allergic asthma (potentially eligible for anti-IgE therapy), while it was indicated that 39.5% of them would also respond to antieosinophilic therapy (anti-IL-5(R) therapy). From 40.6% of eosinophilic asthma patients (probable responders to anti-IL-5(R) therapy), 75.8% could potentially respond to anti-IgE treatment due to their allergic reactivity [154]. Moreover, all of these measurements are unstable over time, and thus, it is uneasy to establish a simple and generalizable interpretation method from a single laboratory examination [155].

Taken together, even though we regularly assess asthma patients using all validated biomarkers, the general clinical nature of the disease (clinical phenotype) is always a blend of distinct processes (inflammatory endotypes) that are potentially susceptible to more plausible interventions (pharmacological or non-pharmacological) [156]. This redundancy may sometimes cause disappointment due to a lack of efficacy, since targeting one distinct immunopathological process with a biological cure may be insufficient. Therefore, it is not surprising that biological treatment, in general, has the potential to reduce severe exacerbations by only 30–50% [157]. Antieosinophil treatment targeted at IL-5(R) in severe eosinophilic asthma patients has a response rate of roughly 83% (69% responders and only 14% super-responders defined by clinical, laboratory and spirometry improvements) [158]. Although the clinical and laboratory characteristics of patients who are potentially eligible for anticytokine (anti-IL-5) or antireceptor (anti-IL-5R or anti-Siglec-8) treatments are virtually the same, the ramifications of these approaches differ significantly, including the diverse nature of groundbreaking exacerbations (a clinical characteristic of imminent treatment failure) [57,159] requiring distinct management [160]. Further, only 30.12% of asthma patients on mepolizumab and 40% on benralizumab experience complete disease remission after 12 months of treatment [161,162,163].

All of the abovementioned limitations of successful severe asthma biological therapy necessitate a more detailed understanding of the immunopathology of the disease. We believe that an improvement in treatment outcomes will rely on obtaining a better understanding of disease biology, leading to more precise diagnostics and endo/phenotyping of the disease. The currently validated biomarkers (AEC, FeNO and IgE) are insufficient to unveil super-responders fromall severe asthma patients and thus give only a blurred picture of the disease phenotype. On the other hand, there is emerging evidence that an increased number of hypodense/activated eosinophils in the peripheral blood is a pathognomic feature of asthma [147] that correlates with both an antigen challenge of the airways [136] and with disease severity [137]. Eosinophil phenotype shifts [143] are probably in line with the proposed dysregulation of eosinophil subsets (rEos/iEos) [5].

Moreover, the respective cellular populations may be readily detectable in the peripheral blood. The dysregulation of their rates can reflect allergen challenges in allergic asthma patients [149] or the presence of comorbidities, such as CRSwNP [140]. It also seems conceivable that the assessment of differences in representations of respective eosinophil subsets, associated with a different dependency of eosinophils on IL-5 and potentially connected with a supposed distinct interference with Type-2 signalling pathways, may provide invaluable information. Hopefully, these measurements will significantly improve and reinforce our decision trees for specific biological treatments, thus effectively extending our current diagnostic opportunities and consequently strengthening disease control and patient quality of life [6]. Further clinical data supporting this hypothesis are anticipated with high hope.

## 9. Conclusions

Eosinophils represent a fascinating cell population with a vast field of action in the human body. Our understanding of their roles in specific pathologies has exponentially increased over the last 20 years, which has led to the development of targeted antieosinophil biological therapies based upon interactions with either the major eosinophil growth factor (IL-5) or with eosinophil surface molecules (IL-5Rα or Siglec-8) to induce cell death through apoptosis. Both approaches represent treatment modalities with significant clinical or laboratory ramifications and differences. Although examination of the eosinophil count in peripheral blood has become a condition that is “*sine qua non*” in asthma diagnostics and phenotyping, the results simplify complex biological processes under the asthma umbrella. Unfortunately, this isolated information is insufficient to render a clear picture of the disease and ease the decision about which treatment approach has a greater chance of successfully achieving control of a patient’s asthma or even inducing clinical or complete remission. This unsatisfactory situation is a tremendous challenge in the search for more detailed diagnostic approaches based on the complex clinical background associated with a repetitive biomarker analysis. We suggest one possible extension regarding an eosinophil immunophenotype analysis that may provide helpful information for the future.

## Figures and Tables

**Figure 1 ijms-24-05716-f001:**
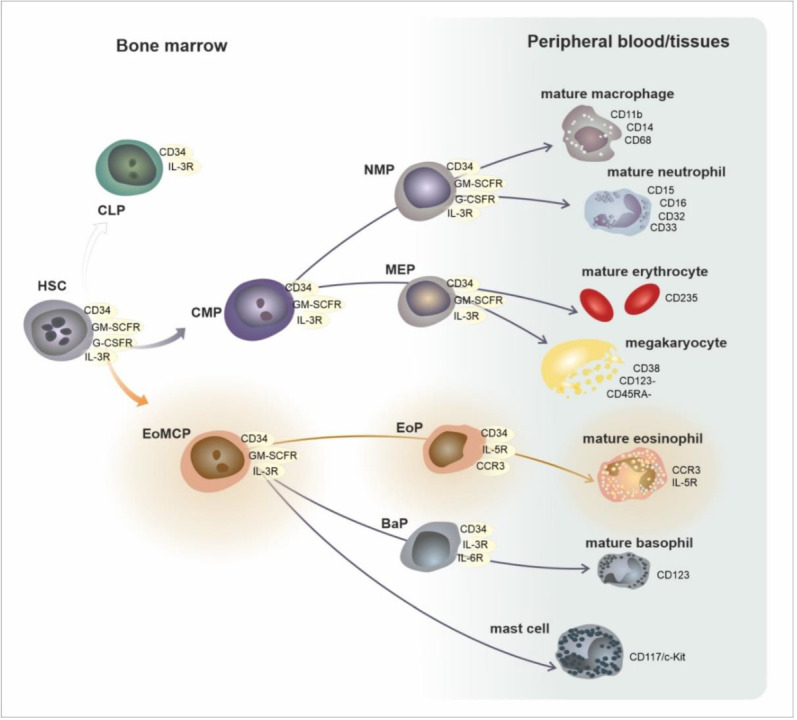
Development of eosinophils in the framework of general hematopoiesis. Abbreviations: HSC—hematopoietic stem cell, CLP—common lymphoid progenitor, CMP—common myeloid progenitor, EoMCP—eosinophil and mast cell progenitor, NMP—neutrophil and macrophage progenitor, MEP—megakaryocyte and erythrocyte progenitor.

**Figure 2 ijms-24-05716-f002:**
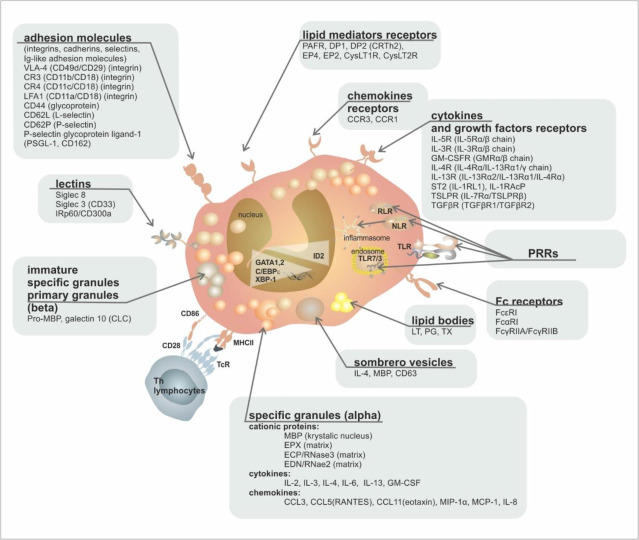
Eosinophil ultrastructure. Abbreviations: MBP—major basic protein, EPX—eosinophil peroxidase, ECP—eosinophil cationic protein, EDN—eosinophil-derived neurotoxin, PRRs—patterns recognising receptors, TLR—Toll-like receptor, RLR—RIG (retinoic acid-inducible gene-I)-like receptor, NLR—NOD (nucleotide-binding oligomerization domain)-like receptor, LT—leukotriene, PG—prostaglandin, TX—thromboxane, DP—D-Prostaglandin receptor, EP—E prostaglandin receptor, pro-MBP—preform of major basic protein, CLC—Charcot–Leyden crystals.

**Figure 3 ijms-24-05716-f003:**
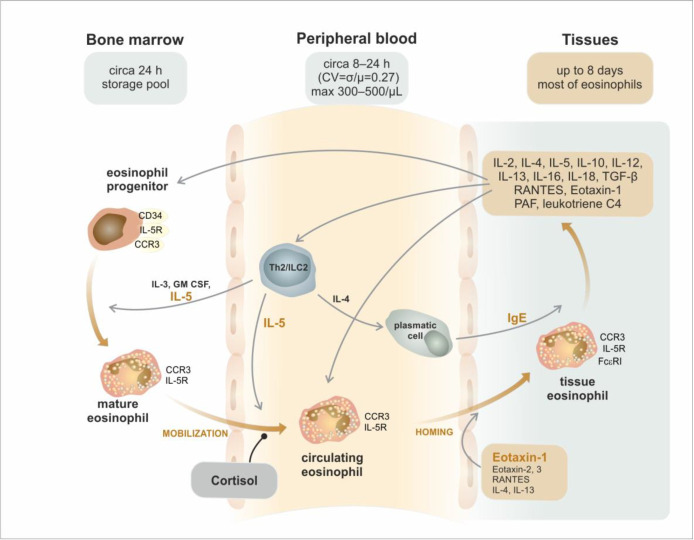
Eosinophil trafficking from the bone marrow to the periphery. After the maturation of eosinophils in the bone marrow, mature cells move into the peripheral blood. The half-life of eosinophils in the circulation is relatively short (approx. 8–24 h) with significant variability (Coefficient of Variation is 0.27), and the average count does not exceed 300–500 cells/μL, depending on the endogenous production of cortisol. Then, the eosinophils settle in peripheral tissues in reaction to eosinophil adhesion (rolling, sticking, and emigration) induced by several cytokines and chemokines. Most of these cells survive and produce highly effective substances in the tissue environment and thus influence the close surroundings and further eosinophil production in the bone marrow as a back-loop.

**Figure 4 ijms-24-05716-f004:**
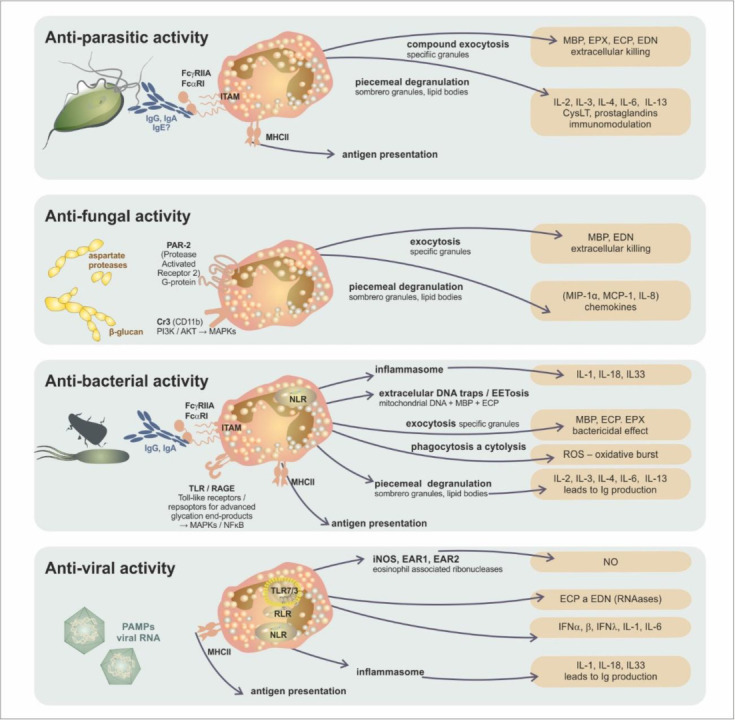
Antimicrobial functions of eosinophils. Eosinophils are multipotent defence cells that provide protective immunity against parasitic, fungal, bacterial, and viral infections. They can interact with microbial bodies, inactivating or killing them in several ways. These features are dependent upon the unique mechanism of protein secretion employed. For a more detailed explanation of the mentioned mechanisms, see Table 3.

**Figure 5 ijms-24-05716-f005:**
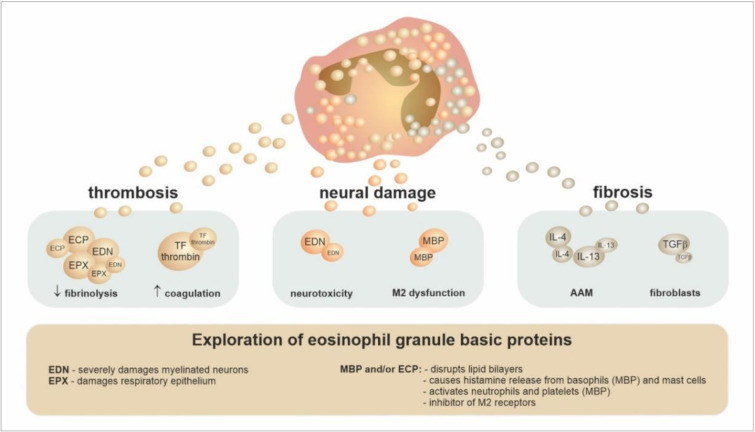
Role of eosinophils in the tissue pathology. The eosinophil infiltration of tissues during various eosinophil-associated diseases is inherently associated with the detrimental effect of the abundant production of several proteins and cytokines. These molecules are subsequently responsible for activating hemostasis (by the damping fibrinolysis and strengthening coagulation) and creating thrombi concurrently with neural damage and fibroproduction. All of these processes are included in the complex tissue pathology. Abbreviations: MBP—major basic protein, EPX—eosinophil peroxidase, ECP—eosinophil cationic protein, EDN—eosinophil-derived neurotoxin, TF—tissue factor, M2 receptor—type 2 muscarinic receptor, AAM—alternatively activated macrophages.

**Figure 6 ijms-24-05716-f006:**
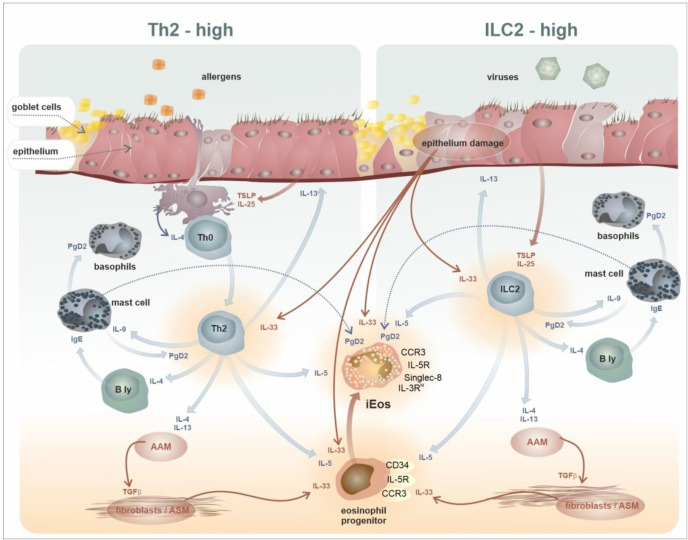
Central role of eosinophils in the complex Type 2-high immune reaction. The complex immunopathology in type 2-high inflammation involves many distinct cell types from the immune system and the tissue interstitium and epithelium. All of these cells can react to and produce signalling molecules, such as cytokines or prostaglandins. The eosinophils (either as progenitors or as mature—proinflammatory cells) are at the core of these processes. The eosinophils particularly respond to IL-5 but also to PgD2, eotaxins, or IL-33. The primary sources of IL-5 are Th2 or ILC2 cells. Th2 cells orchestrate antigen-specific allergic reactivity (specific immune reaction). On the contrary, ILC2 cells are activated independently on specific antigens (innate immune response). Both signalling pathways (Th2-high and ILC2-high) represent, in general, an axis symmetry. Abbreviations: Th2—T helper cells, subset 2, ILC2—Innate Lymphoid Cells, type 2, B ly—B lymphocytes, AAM—Alternatively Activated Macrophages, ASM—Airway Smooth Muscles, PgD2—Prostaglandin D2, iEos—inflammatory eosinophils.

**Figure 7 ijms-24-05716-f007:**
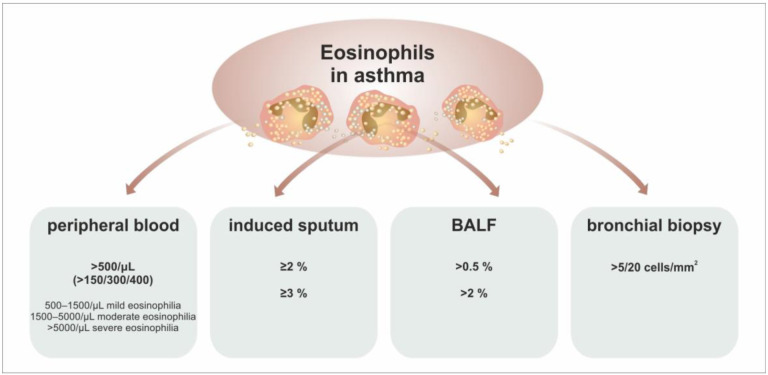
Evaluation of eosinophils and cut-off levels for different biological materials. The presence of eosinophils in various biological samples collected from asthma patients represents a critical characteristic for the definition of type 2-high (or eosinophilic) inflammation. However, the cut-off levels in most of these materials are not unified across the literature. In the peripheral blood, the normal AEC is 500 cells/μL. However, according to the results of clinical studies, much lower counts (150/300/400) are associated with the clinical efficacy of antieosinophil therapy. In the induced sputum, the most commonly cited cut-off level is 2–3%, and in BALF, it is 0.5–2% of all analyzed cells. In bronchial biopsy samples, there is a wide range of cut-off levels in the literature (5–20 cells/mm^2^). Different levels of method validation or standardization may cause these differences. Abbreviations: AEC—Absolute Eosinophil Count, BALF—Brocho-alveolar Lavage Fluid).

**Figure 8 ijms-24-05716-f008:**
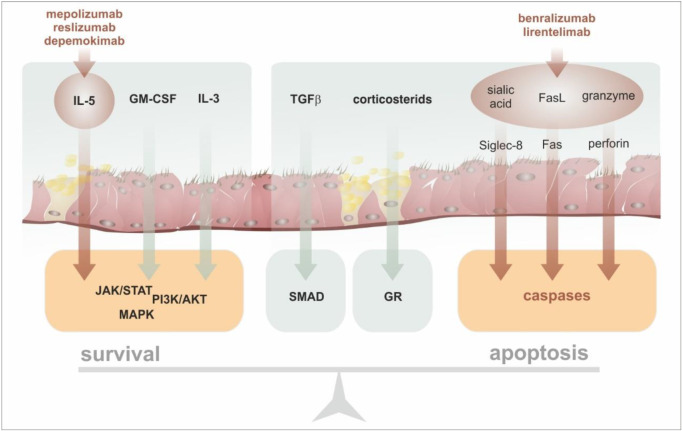
Mode of action of antieosinophil biologicals. Eosinophil survival is always a result of the balance between prosurvival and proapoptotic signals. In general, mepolizumab, reslizumab, and depemokimab are biologicals with an anti-IL-5 mode of action that lead to a reduction in the lifespan of eosinophils. In contrast, benralizumab can induce the apoptosis of targeted eosinophils by utilizing ADCC through ligation of the Fas receptor and the activation of caspases in response to an intracellular influx of granzyme. Lirentelimab can signal through the inhibitory Siglec-8 receptor. Abbreviations: Fas/FasL (CD95/CD95L)—subgroup of the tumour necrosis factor receptor family that contains an intracellular ‘death domain’ and can trigger apoptosis, GR—glucocorticosteroid receptor, TGFβ—transforming growth factor β, SMAD/JAK/STAT/PI3K/MAPK—intracellular kinases.

**Figure 9 ijms-24-05716-f009:**
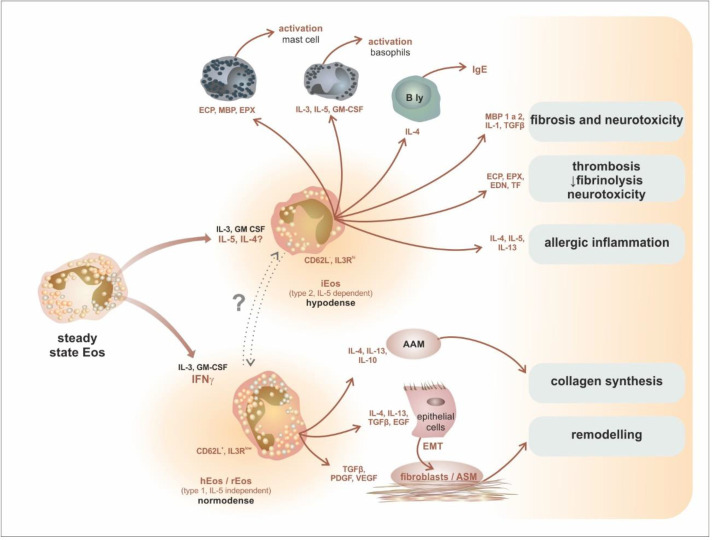
Complex role of eosinophils in the tissue through the prism of their subsets. The current view of eosinophil subsets is in accordance with Abdala Valencia’s concept. Steady-state eosinophils can differentiate into proinflammatory eosinophils (iEos, IL-5-dependent, similar to hypodense eosinophils) and regulatory/homeostatic eosinophils (hEos/rEos, IL-5-independent, similar to normodense eosinophils). Both cell types have distinct phenotypes and functional characteristics, leading to different consequences of immune/tissue reactions. iEos are involved in allergic inflammation, the activation of mast cells, basophils, and B-lymphocytes, fibrosis, neurotoxicity, and coagulation. hEos probably stimulates collagen synthesis initiated by AAM and possibly other reparatory processes, such as remodelling or EMT. Abbreviations: AAM—Alternatively Activated Macrophages, ASM—Airway Smooth Muscle cells, EMT—Epithelial-Mesenchymal Transition.

**Table 1 ijms-24-05716-t001:** Development of eosinophils.

Area/Subset	Developmental Cell Type	T1/2	Phenotype
**bone marrow**	**CMP**—common myeloid progenitor**EoMCP**—eosinophil/mast cell progenitor	circa 24 h	CD34, GM-SCFR, IL-3R

	**EoP**—eosinophil progenitor	CD34, GM-SCFR, IL-3R, CCR3, IL-5Rα
	**mature eosinophil**	GM-SCFR, IL-3R, CCR3, IL-5Rα, VLA-4 (CD49d/CD29), PSGL-1 (CD162)
**peripheral blood**	**mature eosinophil**	8–24 h	GM-SCFR, IL-3R, CCR3, IL-5Rα, VLA-4 (CD49d/CD29), PSGL-1 (CD162)
**peripheral tissues**	**mature eosinophil**	8 days	GM-SCFR, IL-3R, CCR3, IL-5Rα, FcεRI, VLA-4 (CD49d/CD29), PSGL-1 (CD162)
	**activated eosinophil**	?	GM-SCFR, IL-3R, CCR3, IL-5Rα, FcεRI, CD63, CD9, CD69
**Mature eosinophil subsets in mice and humans**
**mice**	**hEos**—homeostatic eosinophil**rEos**—resident eosinophil (IL-5 independent)	lungs: 36 h, GIT: 6 days	Siglec-F^med^CD62L^+^ CD101^lo^, CCR3, IL-5R, F4/80, CD11c^−^
	**iEos**—inflammatory eosinophil (IL-5 dependent)	?	Siglec-F^hi^CD62L^−^CD101^hi^, CCR3, IL-5R, CD11c^low^
**humans**	**hEos**—homeostatic eosinophil**rEos**—resident eosinophil (IL-5 independent)	?	Siglec-8^+^CD62L^+^IL-3R^low^
	**iEos**—inflammatory eosinophil (IL-5 dependent)	?	Siglec-8^+^CD62L^low^IL-3R^high^

**Table 2 ijms-24-05716-t002:** Eosinophil receptors.

Group of Receptors	Receptor	Ligand	Function
**cytokine and growth factor receptors**	**IL-5R** (IL-5Rα/β chain)/(CD125/CD131)	IL-5	proliferation, growth, bone marrow escape, survival
	**IL-3R** (IL-3Rα/β chain)/(CD123/CD131)	IL-3	growth, survival
	**GM-CSFR** (GMRα/β chain)/(CD116/CD131)	GM-CSF	growth, survival
	**IL-4R** (IL-4Rα/IL-13Rα1/γ chain)	IL-4	activation
	**IL-13R** (IL-13Rα2/IL-13Rα1/IL-4Rα)	IL-13	activation
	**ST2** (IL-1RL1)	IL-33	growth, survival
	**TSLPR** (IL-7Rα/TSLPRβ)	TSLP	growth, survival
	**TGFβR** (TGFβR1/TGFβR2)	TGFβ	inhibition of survival
**chemokine receptors**	**CCR3**	Eotaxin 1, 2, 3 (CCL11/CCL24/CCL26), MCP-3, MCP-4	chemotaxis
	**CCR1**	MIP-1α (CCL3), RANTES (CCL5)	chemotaxis
**lipid mediator receptors**	**PAFR**	PAF	activation
	**DP2 (CRTh2)**	PgD2	chemotaxis
	**DP1**	PgD1	chemotaxis
	**EP4**	PgE2	activation/inactivation
	**EP2**	PgE2	activation/inactivation
	**CysLT1R**	LTD4, LTC4, LTE4	activation
	**CysLT2R**	LTD4, LTC4, LTE4	activation
**Pattern Recognition Receptors (PRRs)**	**TLR1, 2, 3, 4, 5, 6, 7, 9, 10** (Toll-Like Receptors)	PAMPs	activation, degranulation
	**NLR1, 2** (NOD-Like Receptors)	PAMPs, DAMPs	activation
	**RLR** (RIG-Like Receptors)	dsRNA	activation
	**PAR-2**	protease-activated receptor	activation
	**RAGE** (Receptor for Advanced Glycation Products)	Advanced Glycation Products	activation
**Fc receptors**	**FcεRI**	IgE	nonactivation
	**FcαRI**	IgA	activation
	**FcγRIIA/FcγRIIB**	IgG	activation/inactivation
**MHC**	**MHCII** (+CD80, +CD86, +CD40)	TCR + CD4	antigen presentation
**adhesion molecules** (integrins, cadherins, selectins, Ig-like adhesion molecules)	**VLA-4** (CD49d/CD29) (integrin)	VCAM, fibronectin	activation, adhesion
	**CR3** (CD11b/CD18) (integrin)	iC3b	activation, adhesion
	**CR4** (CD11c/CD18) (integrin)	iC3b	activation, adhesion
	**LFA1** (CD11a/CD18) (integrin)	ICAM-1, ICAM-2	activation, adhesion
	**CD44** (glycoprotein)	hyaluronic acid	adhesion, homing
	**CD62L** (L-selectin)	CD34, GlyCAM-1, MadCAM-1	adhesion, homing
	**CD62P** (P-selectin)	P-selectin glycoprotein ligand-1 (PSGL-1)	activation, adhesion
	**PSGL-1** (P-selectin glycoprotein ligand, CD162)	P-selectin (CD62P)	activation, adhesion
	**CD34** (fosfoglykoprotein)	L-selectin	adhesion, migration
**lectins**	**Siglec-8**	Sialyl–Lewis X (CD15s)	apoptosis induction
	**Siglec-3 **(CD33)	sialyl acid	apoptosis induction
	**IRp60/CD300a**	sialyl acid	inhibition of growth signals (IL-3/IL-5/GMCSF)

**Table 3 ijms-24-05716-t003:** Antimicrobial roles of eosinophils.

Antimicrobial Immunity	Antigen Recognition by Eosinophils	Innate Immune Reaction Mediated by Eosinophils	Eosinophil-Dependent Adapted Immune Reaction
**antiparasitic**	IgA and IgG bonded to the parasite surface/FcαRI, FcγRII	cytotoxic proteins (MBP, EPX, ECP, EDN)	parasite antigen presentation
			T-cell modulation response toward the Th2 subset
			stimulation of IgM production
**antifungal**	mould aspartate proteases/PAR-2 (Protease Activated Receptor 2)	cytotoxic proteins (MBP, EDN)	unclear
	β-glucan/CR3 (CD11b)	chemokine production (MIP-1α, MCP-1, IL-8)	
**antibacterial**	PAMPs PRRs (TLR/Toll-like receptors/, NLR/NOD-like receptors/, RAGE/Receptor for Advanced Glycation Endproducts/)	phagocytosis and cytolysis (MBP, ECP)	bacterial antigen presentation IgM production stimulation
		bactericidal effect (MBP, ECP, EPX, production of ROS)	
		extracellular traps (mitochondrial DNA, cytotoxic proteins)	
**antiviral**	PAMPs	EDN and ECP—RNAase activity	viral antigen presentation
	PRR (TLR7/3, RLR/RIG-I-like receptor/, NLR/NOD-like receptors/)	iNOS induction, NO synthesis	T-cell response modulation toward the Th1 subset
		IL-6 production	

**Table 4 ijms-24-05716-t004:** Eosinophils in asthma.

The Cut-Off Level of AEC	Frequency in Asthma	Clinical Interpretation
>150/μL	69%	T2-high phenotype according to the GINA recommendations, prediction of the clinical response to anti-IL-4R therapy
>300/μL	37%	prediction of the clinical response to antieosinophil treatment (anti-IL-5 (R))
		probable sputum eosinophilia
		not significantly predictive concerning anti-IgE and anti-TSLP therapy
>400/μL	16%	high risk of loss of control of asthma

**Table 5 ijms-24-05716-t005:** Antieosinophil therapy for asthma patients.

Feature		Mepolizumab (anti IL-5), IgG1 kappa	Reslizumab (anti IL-5), IgG4 kappa	Benralizumab (anti Il-5Rα), IgG1 kappa
**laboratory**	**baseline AEC**	≥150/μL at baseline or ≥300/μL last year	≥400/μL	not limited initially, posthoc ≥ 300/μL vs. <300/μL
	**allergy, baseline IgE**	no limitation	no limitation	no limitation
	**baseline FeNO**	no limitation	no limitation	no limitation
**clinical**	**severe asthma exacerbation rate**	≥2	≥1	≥2
	**inhaled corticosteroids**	high doses	medium to high doses (≥440 μg fluticasone/day)	high doses
	**oral corticosteroids**	no limitation	limited ≤ 10 mg prednisone/day	no limitation
	**FEV1 (%predicted)**	<80% (<90% for age <18 yrs) at screening	no limitation	<80% (<90% for age <18 yrs) at screening and randomisation
	**ACQ score**	no limitation	ACQ-7 ≥ 1.5 at screening and randomisation	ACQ-6 ≥ 1.5 at screening
	**GINA treatment step**	4–5	3–5	4–5
**dosing**	**dosing interval**	4 weeks	4 weeks	4/8 weeks
	**dose**	100 mg (300 mg EGPA)	3 mg/kg iv for 20–50 min	30 mg
	**route of application**	sc	iv	sc
	**home-use**	yes	no	yes
**approved**	**other indications**	EGPA, HES, CRSwNP	none	none

**Table 6 ijms-24-05716-t006:** Haematological effects of antieosinophil therapy.

	Mepolizumab	Reslizumab	Benralizumab
**Dosing/indication**	100 mg/4 weeks sc	3 mg/kg/4 weeks iv	30 mg/(4)8 weeks sc
	(**asthma, CRSwNP, EGPA, HES**)	(**asthma**)	(**asthma**)
	300 mg/4 weeks sc (EGPA)		
	300 mg sc/750 mg iv/4 weeks (**HES**)		
**The onset of the clin. effect**	24 h.—max. in 4 weeks	24 h.—max.?	24 h. max.
**Eosinophils BM**	**↓ 70% (elevation of progenitors)**	**↓ ?**	**↓ ~100% (incl. progenitors)**
**Eosinophils PB**	**↓ ~76–88%** (according to dosing)	**↓ ~77–92%**	**↓ ~100%**
**Eosinophils in sputum**	**↓ 50% (progenitors?)**	**↓ circa 56%** (vs placebo)	**↓ 82–96%**
**Eosinophil receptors**	**↑ mIL-5Rα?, ↓ IL-3Rα, ↑CRTH2**	IL-5Rα?	**↓ IL-5Rα mRNA**
	CCR3 no change		
**Other cells**	**No effect on basophils**, mast cells?, no effect on T-ly or the **elevation of T-ly IL-5+**	No effect on T-ly	**↓ basophils**, mast cells? ILC2?
**Other cytokines**	**↑ IL-5, CCL-13, CCL-17, CCL-22,**	**↑ IL-5**, no change in IL-3	IL-5?, IL-3?, **↑ eotaxin 1 and 2, IL-6**
	**eotaxin 1 (CCL-11),** IL-3?		

↑-heightening, ↓-lowering.

## Data Availability

Not applicable.

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
