# Peer review of "Subsets of Eosinophils in Asthma, a Challenge for Precise Treatment"

_ijms, 2023, doi:10.3390/ijms24065716_

Round 1
Reviewer 1 Report
The review "Subsets of eosinophils in bronchial asthma, a challenge for precise treatment" is timely, competently written, covering both eosinophil biology and clinical aspects of eosinophilic asthma. The text is easy to read, new literature is used. The review will be useful for anyone interested in eosinophils and their roles in respiratory medicine.
Minor revision:
1) There are some grammatical errors in Table 5: ciorticosteroids, FEV1 (% n.h.), fluticasona/day), <18 let
2) Unify terminology: corticoid, glucocorticoids, corticosteroids, steroids, corticotherapy.
3) The term "bronchial asthma" is somewhat outdated, less often used, I suggest that the authors could discuss the use of the term "asthma".
Reviewer 2 Report
Novosad et al have carefully written a thorough and nice review on eosinophils (Eos). They have covered Eos development, differentiation, function, phenotypes in health and disease (especially in asthma), used biomarkers and biological mAb currently in clinical use to inhibit their unwanted effects. This review gives a complete and holistic view of the knowledge on Eos now and is leaning to large amount (163) references. The authors also bring up also unknown or unclear points, which can serve ideas for the future studies to come.
General comments
1. The weakness of this review: the figures. The idea in the figures is good and supported but they have very bad resolution, small text (sometimes even impossible to read) and garish colors (too much contrast). Also, the same cell looks different in different figures. I highly recommend using BioRender or some other similar program designed to help figure design. Also, the background of the figures is taking too much attention (different colors, or gray). Why not have white background if no reason to indicate something special with the background color?
2. The figures would be much more informative, if they have a short description what is happening in the figure or what it is showing (more than just a title). Also in the reviews, the figures should be self-explanatory (without reading the full text).
3. Also, the tables (especially Tables 1-3) are a bit messy. Try to make them more reader-friendly, e.g. by adding some vertical lines between the columns and widening columns with only a little text, and broadening the other columns with large amount of text.
4. The font in the tables and figures is different from the text (maybe courier and Times, respectively?) and gives unpolished impression.
5. The review is quite long, shortening would crystallize the message. Often the sentences could have said much shorter, there’s also some repetition.
Specific comments
1. Table 1: Subsets of Eos don't fit in the first column under title "area"! Modify either the first column title (“area”) so that it covers both different locations and different subsets, or reorganize the table some other way. Now it’s not logical.
2. Table 2: Stylish also this table. The column "group of receptors" is too wide -> more space needed for column "receptor".
3. Table 2: typo: chemokines repetozs -> chemokine receptors
4. End of page 8, lines 222-223: Although their activation and degranulation may also occur [34]. -> This sentence seems somehow incomplete. Either begin with “However,”, or modify some other way to fulfil.
5. Why there are so many "/../" marks? -> Could you replace those with "," or something else, better?
6. In Figure 3 you say that Eos circulate in the peripheral blood ca 8-24h, but in the text (page 8) you say that their half-life is only 3h in the blood. How this should be understood? Please clarify.
7. Table 3 referred after the actual table. Please, first text + reference to table/figure, and after that the presentation.
8. Figure 7, What is the difference between 2% or 3% in induced sputum, or >0.5% and >2% in BALF? The figure needs some additional explanation.
9. Figure 9 referred after the actual figure. Please, first text + reference to table/figure, and after that the presentation. Also additional explanation needed for this fig.
10. Line 605: correct “Tab. 5.” To “Table 5.”
